# Watching Television While Eating Increases Food Intake: A Systematic Review and Meta-Analysis of Experimental Studies

**DOI:** 10.3390/nu17010166

**Published:** 2025-01-02

**Authors:** Dresshti Garg, Evelyn Smith, Tuki Attuquayefio

**Affiliations:** 1School of Psychology, Western Sydney University, Sydney, NSW 2751, Australia; e.garg@westernsydney.edu.au (D.G.); evelyn.smith@westernsydney.edu.au (E.S.); 2Translational Health Research Institute, Western Sydney University, Sydney, NSW 2751, Australia

**Keywords:** television, television viewing, eating, intake, obesity, distraction, memory, attention

## Abstract

**Background/Objectives**: Television viewing has been linked with increased weight and obesity, likely through decreased physical activity associated with sitting and viewing television, as well as increased intake of food, likely through reduced awareness of eating and intake behaviours. This review sought to determine the effects of television viewing on energy intake relative to the absence of television. **Methods**: We adhered to the PRISMA guidelines and pre-registered this review in PROSPERO (CRD42023493092). The PICOS strategy included children, adolescents and adults of all ages (P), exposed to television viewing only during meals (I) compared to no television and no other distractors (C), with the outcome as energy intake or consumption (O) for both within-subject and between-subject randomised controlled trial (RCT) designs (S). **Results**: Robust-variance meta-analyses of k = 57 effect sizes from 23 studies showed no overall effect, noting high heterogeneity. When analyses were limited to television alone with k = 29 effect sizes from 23 studies, we revealed a small significant effect of television viewing on intake (*g* = 0.13, 95% CI [0.03–0.24]) compared to no television. Moderation analysis showed that television viewing strongly increased intake at the next meal (*g* = 0.30, 95% CI [0.03–0.57]) but not immediate intake (*g* = 0.10, 95% CI [−0.01–0.21]). **Conclusions**: This review showed that television viewing increases food intake, especially at the next meal. This effect was evident across both children and adults. This review highlights how television viewing impacts intake and offers potential avenues for intervention based on our findings.

## 1. Introduction

Obesity is a worldwide health problem, and the rate of obesity continues to rise to 58% [1], particularly in children and adolescents. One potential contributor to weight gain is the presence of distracting stimuli during eating bouts, such as televisions, phones, iPads, video games, reading, music and other individuals. There is a wealth of cross-sectional studies showing that viewing TV while eating is associated with higher food consumption, particularly unhealthy options. The interaction between TV viewing and eating behaviours, therefore, has implications for health and dietary choices. Multiple recent reviews of experimental studies indicate that the presence of such distractors increases energy intake [2,3,4].

Television viewing has become a significant part of daily modern life. Increased television viewing increases sedentary behaviour and reduces time spent undertaking physical activity, leading to weight gain [5]. Observational studies show that television viewing during mealtimes is associated with poorer dietary quality, including higher intakes of sugar-sweetened beverages and energy-dense, nutrient-poor foods [6]. One study found that most people eat while watching TV, and this leads to longer eating durations and increased total eating time [7]. There are also multiple studies indicating that children are particularly vulnerable to the effects of TV on their eating behaviours. Two systematic reviews of cross-sectional studies have highlighted the inverse association between mealtime TV use and diet quality in children and adolescents [6,8]. Additionally, children who watch more TV are more likely to consume sugary foods, which can lead to health issues such as dental caries [9]. Longitudinal studies indicate that early exposure to television correlates with increased unhealthy food consumption patterns later in life, particularly among young adults [10].

Given the ubiquity of televisions in everyday life and their increasing presence at mealtimes [11], the investigation of its influence on food intake is warranted. While the impact of multiple distractors, including television, reading, music, phones, iPads, and video games, has been explored in previous systematic reviews [3] and meta-analyses [2,12], and these reviews highlight the multi-faceted and potentially cumulative impacts of multiple types and combinations of distractions on energy intake; the specific impact of television viewing on eating behaviour remains unclear. While one review examined the effects of television viewing on energy intake in children and adolescents [12], only four studies were eligible for meta-analytic review. Further, the limit to children and adolescents missed key studies in adults picked up in other reviews [2], leading to a disconnected literature synthesis. As such, the current review sought to examine the impact of television viewing on energy intake across the lifespan (i.e., in children, adolescents, and adults of all ages) in experimental studies. To date, no review has examined the effects of television viewing compared to no television in experimental studies exclusively. The focus on experimental studies allowed for closer examination and provided a more accurate picture of the causal impact of television viewing on eating behaviour in a well-controlled environment, minimising confounding variables. Finally, we explored whether the impact of television viewing on energy intake was moderated by common external factors (e.g., age, sex, body mass index, design, time of intake).

## 2. Materials and Methods

### 2.1. Protocol and Pre-Registration

The protocol for this review was registered on 19 December 2023 via the International Prospective Register of Systematic Reviews (PROSPERO), ID: CRD42023493092. The protocol was based on Preferred Reporting Items for Systematic Reviews and Meta-Analysis (PRISMA).

### 2.2. Eligibility Criteria

The following inclusion criteria were applied: experimental studies or randomised controlled trials comparing television viewing to no television (or other distractor), measure energy intake as the primary outcome, published between 1990 and 2024, published in a peer-reviewed journal, in the English language, and have full text available. The following were excluded: self-report, questionnaires/surveys on television viewing, systematic, narrative or scoping reviews, qualitative analysis, observational studies, and cross-sectional design, studies with non-human participants. Exclusion was not made based on age, sex, and healthy or unhealthy food intake. The research question encompassed the PICOS criteria: Participants: all age groups across both sexes, Intervention: television watching, Comparison: absence of television and any other distractor, Outcome: energy intake or volume, Study type: experimental studies with random allocation to intervention or control groups.

### 2.3. Search Strategy

Information was extracted from four online databases: PsycINFO, EMBASE, Web of Science and PubMed. The last search was conducted on 8th January 2024. The search strategy was generated from a combination of keywords in papers, key concepts in the field and strategies used in other reviews [3,4,12]. The following terms were included: eating behaviour, feeding behaviour, food intake, calorie intake, nutrient intake, dietary intake, meal, snacking, portion size, overeating, television, TV, screen, and distract. The full search strategy can be found in Appendix A.

### 2.4. Study Screening

Studies identified via electronic sources were imported into Covidence for title, abstract and full-text screening. Two assessors (TA and DG) screened potential articles independently at every stage with 98% agreement. The studies were first screened based on their title and abstract, and all conflicts were resolved by the collaborative understanding of the two researchers (TA and DG). Finally, 103 were full-text screened, of which 25 studies were identified and eligible for review from the electronic searches. A manual search using backward chaining (428) and forward chaining (1266) was run to supplement the electronic searches. From this, an additional 4 studies were deemed eligible, leaving 29 studies for systematic review (see the PRISMA flowchart in Figure 1 for a full breakdown). One publication [13] includes two experimental studies (i.e., there are twenty-nine studies from twenty-eight individual publications).

### 2.5. Data Extraction

Primary outcomes were means and standard deviation from experimental (TV) and control (No TV) conditions. Means and standard deviations were estimated using sample size, median, range and/or confidence intervals [14]. Authors were also contacted to request additional data not otherwise reported. If none of these statistics were reported or available, the study was excluded from the meta-analysis. Six studies did not include sufficient information to be included in the meta-analysis. This left 23 studies available for meta-analysis.

Means and standard deviations were used to calculate standardised mean differences and converted to Hedge’s g as a more unbiased estimate of the population standardised mean difference. Individual and pooled effect sizes and confidence intervals were illustrated using forest plots, where a positive effect size reflected increased energy intake while viewing television (relative to no television viewing).

### 2.6. Meta-Analysis

Data were analysed using the ‘meta’ package (version 7.0.0) [15] in R (R Core Team, 2017). Due to variation in samples and methodology, we used a random-effects model (with a restricted maximum-likelihood estimator) to account for the variation in the true effects. As several studies compared multiple intervention groups to common control, we performed the meta-analytic technique of robust variance estimation [16,17]. Under this model, we assumed correlated weights and ρ set to the recommended 0.80. Z-scores of standard residuals exceeding 1.96 were used to determine outliers using the grubbs.test() function in the ‘outliers’ package. Heterogeneity was assessed using the *τ*^2^, *I*^2^, and *Q* statistics. *τ*^2^, estimated using the restricted maximum-likelihood [18], is the variance of the effect size parameters across the population of studies, and it reflects the variance of the true effect sizes. *I*^2^ values indicate the proportion of variability in effect sizes that can be attributed to systematic, between-study differences rather than a within-study error [19]. Q statistics and their associated *p* values test whether such a between-study difference is significant. Finally, a prediction interval for the true outcomes is also provided [20].

### 2.7. Moderator Analyses

We also conducted meta-regressions to determine the impact of key moderators such as percent of females, age, BMI, design, risk of bias and time of intake (immediate or delayed). To interpret significant moderators, we ran follow-up meta-analyses within each of the levels of the moderator. For each significant moderator, we conducted a subgroup analysis using a mixed-effects model, which generated estimates that represented the change in effect size strength with every increasing unit (i.e., age in years).

### 2.8. Small Study and Publication Bias

To identify potential small study bias across our analyses, we conducted Egger’s test of the regression intercept [21] and visually inspected normal and counter-enhanced funnel plots; when Egger’s tests were significant, we provided additional estimates of the effects after adjustments using trim and fill techniques [22].

### 2.9. Risk of Bias Assessment

The risk of bias was assessed by one reviewer (T.A.) using a standardised framework developed by Cochrane. The risk of bias was assessed according to the Cochrane RoB 2 tools for parallel randomised clinical trials (RCT) (https://www.riskofbias.info/welcome/rob-2-0-tool/rob-2-for-cluster-randomized-trials, Accessed on 15 November 2024) and for Crossover trials (https://www.riskofbias.info/welcome/rob-2-0-tool/rob-2-for-crossover-trials, Accessed on 15 November 2024). Studies were classified as either Low, Some Concerns, or High risk of bias across 5 or 6 domains for parallel trials or cross-over trials, respectively, and then given an overall risk of bias.

## 3. Results

### 3.1. Study Design and Sample Size

Several study designs were used in this review; 13 (45%) were within-subject designs [23,24,25,26,27,28,29,30,31,32,33,34], 13 (45%) were between-subject designs [13,35,36,37,38,39,40,41,42,43,44,45,46], and 3 (10%) used a mixed design [47,48,49]. The sample sizes for either intervention or control groups ranged from 8 to 168, with only 1 study including more than 100 participants in a group [36].

### 3.2. Sex

The sex of the participants varied across the review; 11 (38%) studies included 100% female participants [13,25,26,27,30,33,41,42,48,49], 16 (55%) studies had both sexes [23,24,28,29,31,34,35,36,37,38,40,42,43,44,46], and the rest (7%) were described as males only [32,47].

### 3.3. Age Range

The mean age for participants ranged from 4.60 to 31.90, with 17 (59%) studies consisting of young adults as participants (mean age > 18 years) [13,23,25,26,29,30,31,35,40,41,42,43,45,48,49] and 11 (38%) studies consisting of children as participants (mean age < 18 years) [27,28,32,33,34,36,37,38,39,44,47]. The age in the relevant groups was not reported for one study [46].

### 3.4. Food Intake

To measure the food intake, 8 (28%) studies used the unit of grams [23,26,28,30,40,41,43,45], 15 (52%) studies used the unit of kcal [24,27,31,33,34,35,36,37,38,39,42,44,46,48], and the rest (21%) used kilojoules [13,25,29,32,47,48,49]. Only one study used both grams and Kcal to measure the food intake [43]. One study asked participants to consume either a single variety or multiple varieties of foods under television versus no television conditions. Food intake was significantly higher in television conditions, and increasing the variety of foods also increased the intake [49]. In other words, the impact of television viewing on intake was greatest when food variety was low, and because food variety also increased intake, this television effect was diminished for these participants.

### 3.5. Food Intake Timing

Regarding timing of food intake, 24 (83%) studies measured consumption immediately with the television viewing [23,24,25,26,27,28,29,31,32,33,34,36,37,38,39,40,41,42,43,44,46,47,48,49] and 5 (17%) studies measured delayed intake (i.e., at a later meal) [13,30,35,45].

### 3.6. TV and Control Conditions

In the testing conditions, 20 (69%) studies compared television with no television with the participant eating alone [13,23,27,29,30,31,32,33,34,35,39,40,41,43,44,45,47,48,49], 6 (21%) included a group element in their control group, including social interactions [28,42] or in the presence of peers in classrooms [36,37,38,46]. The remaining 3 (10%) used reading [26] or listening to audio [24,25] as the control group.

### 3.7. Studies with Additional Elements or Confounds

Studies in this comparator exposed the participants to five levels of advertisements (no television exposure, television with no ads, one ad, two ads and three ads) and analysed their food intake. Including advertisements with television did not significantly impact food intake in these studies [36,37,38,39,46]. Three of these publications also reported on the interaction between advertisements and toys while watching television—all three found no impact of toy gadgets, advertisements and television viewing on intake in children [36,37,46]. It should be noted that the same sample appears to be reported across two studies—one focusing on children in Mexico [37], and another on children across multiple Latin American countries [46] including Mexico. Given the way the statistics were reported, it is not possible to tease apart the samples to remove this overlap.

Five studies [13,26,44,45,48] also assessed whether television content impacted intake. This included whether the content was funny/engaging/boring, whether it contained food cues, or whether it was repeated or continuous. Chapman et al. (2014), in addition to a standard control versus television comparison, also introduced two extended settings of boring versus engaging television to assess the effect of television content on food intake [26]. Participants in the boring television condition consumed the most food, whereas engaging television participants consumed the least. Likewise, in Experiment 2 by Mittal et al. (2011), the greatest intake was seen in boring and funny TV conditions relative to no TV control [13]. One study also examined the effect of continuous or repeated television segments on participants. Food intake was highest in the continuous television group, arguably due to the continuous allocation of attention, which disrupted food habituation. Repeated segments did not have the same allocation of attention, and habituation was noticeable [44].

While we focused here on television viewing compared to control groups, many studies included other experimental conditions, such as social interactions [29,42], walking [42], reading, computer games, and challenges. One study assessed food intake while watching television, driving, social interaction or eating alone. Only the television viewing group consumed significantly more food than other conditions suggesting that television is a greater distractor than the rest [41].The full breakdown of this information is provided in Table 1 in the qualitative review.

### 3.8. Risk of Bias

Risk of bias assessments are presented for parallel and cross-over trials in Figure 2 and Figure 3, respectively. For the parallel RCTs (Figure 2), 6 of the 12 studies were rated as low risk of bias [36,37,38,39,42,46] while the others were rated with some concerns, and none were rated as a high risk of bias. All parallel RCTs did not include a priori analysis plans or pre-registrations, so domain D5 was rated with some concern across all trials.

For cross-over trials (Figure 3), only three trials were rated as low risk of bias [32,43,48] and all other trials were rated with some concern. Again, lack of information on planned or a priori analyses was rated with some concern across all trials except those ultimately rated as low risk.

## 4. Meta-Analysis

### 4.1. Summary of All Studies

When we consider all studies, a total of *k* = 57 effect sizes from 23 studies were included in the analysis. The observed standardised mean differences (Hedge’s *g*) ranged from −2.45 to 0.92. The estimated average standardised mean difference based on the random-effects model was *g* = 0.10 (95% CI: −0.05 to 0.24). A forest plot showing the observed outcomes and the estimate based on the random-effects model is shown in Figure 4.

### 4.2. Meta-Analysis Without Advertisements or Toys

Several studies included additional factors beyond television manipulation. Specifically, five studies based on a common methodology assessed food intake in a group classroom setting while watching television laden with varying numbers of advertisements, and in some cases, children were given toys. As evident in Figure 5, the effect sizes for these studies are small and varied and add significant heterogeneity. Indeed, there is significant heterogeneity (*I*^2^ = 81%), likely due to these additional confounds. As such, an additional meta-analysis was conducted on studies that did not have advertisement and/or toy manipulations in addition to television manipulation. For this second meta-analysis, a total of *k* = 29 effect sizes from 23 studies were included in the analysis. The observed standardised mean differences (Hedge’s *g*) ranged from −0.48 to 0.66. Grubb’s test for outliers indicated no significant outliers in effect sizes (*G* = 4.65, *p* = 0.26). According to the Q-test, the true outcomes appear to be non-heterogeneous (*Q*(28) = 21.86, *p* < 0.0001, *τ^2^* < 0.0001 (between cluster) and *τ^2^* < 0.0001 (within cluster), *I*^2^ = 0.

The estimated average standardised mean difference based on the random-effects model was *g* = 0.13 (95% CI: 0.03 to 0.24). Now, using studies that manipulated only television viewing, the average outcome differed significantly from zero (*t* = 2.70, *p* = 0.01). In other words, across 23 studies, television viewing significantly increased energy intake. A forest plot showing the observed outcomes for these studies is shown in Figure 5.

### 4.3. Assessment of Small Study Bias

Funnel plots are used to detect bias or systematic heterogeneity in the data. Funnel plots are scatter plots of the effect size (Hedge’s *g*) against the standard error of the study. Visual inspection of the inverted funnel shape in Figure 6 suggests that a small study bias is unlikely. The symmetry in the funnels suggests little evidence of small study bias, while the spread falls within the 95% estimated bounds. A linear regression test showed no evidence of funnel plot asymmetry, *t* = 0.21, *p* = 0.835. As such, the observed effects are unlikely due to small study/reporting bias or chance. A funnel plot of the estimates is shown in Figure 6.

### 4.4. Moderation Analyses

Moderation analyses were conducted on individual predictors, including the percentage of females in the sample, average age, BMI, study design, and time of food intake. Results are presented in Table 2 below.

Three moderators returned significant (i.e., *p* < 0.05) results: timing, study design, and risk of bias. These results show that the effects of television appear stronger at a later meal (i.e., delayed timing) when a between-subjects design was used and when the risk of bias was lowest. The strongest moderation effect was related to the timing of food intake. To investigate this further, we also conducted a subgroup mixed effects analysis comparing immediate to delayed intake. Moderation analysis of intake time (immediate vs. delayed) showed a non-significant moderation effect on intake, *F*(1,27) = −1.84, *p* = 0.08 (although there is evidence of a trend). Specifically, the subgroups analysis showed that the effect size for the 8 studies that used delayed intake was significant (*g* = 0.30, 95% CI: [0.08–0.53], *p* = 0.008) while the average effect size for the 21 studies that used immediate intake was close to significance, (*g* = 0.10, 95% CI: [0.00–0.20], *p* = 0.058). Figure 7 below shows the forest plot of effect sizes, split by intake timing (immediate or delayed).

## 5. Discussion

### 5.1. Overview of Main Findings

This review analysed the influence of television viewing on food intake across 29 experimental studies. This review provides both a qualitative systematic synthesis of the literature and a quantitative meta-analysis of the effect of television watching on food intake under experimental conditions. Most studies assessed immediate food consumption, while a smaller number focused on delayed intake. The studies included participants ranging from children with an average age of 4.6 years old, with sample sizes as low as 8 up to 168 (within an allocated group). Some studies used females only, while others included both sexes. Across the 23 studies and 57 effect sizes considered for meta-analysis, there was no overall significant effect of watching television on food intake. Among these studies there were high levels of heterogeneity which may have contributed to this null effect. When we excluded studies that manipulated television in the context of other distractors such as advertisements and/or toys, 29 effect sizes from 23 studies were meta-analysed. From this meta-analysis on a subset of comparisons, there was a significant overall effect of television watching on food intake, noting that this overall effect was small (*g* = 0.13). Across this subset, there was considerable heterogeneity in the effects of television viewing pm food intake, with one reporting a 37% (~73 kcal) reduction in intake [42] and another reporting 57% (~10 g) increase in intake [41]. Further moderation analyses show that this small effect size appears to be driven by the effects of television on later food intake (*g* = 0.30), while immediate intake (*g* = 0.10) was not significantly impacted (noting a trend to significance). This translates to a significant increase in food intake (up to 22% higher) at the next meal [13,30], while immediate intake increased by a more mod-est amount between 5–15% [25,29,42]. This result is surprising as multiple cross-sectional and longitudinal studies show a positive relationship between television viewing time and obesity [5], energy intake [7], and intake of “discretionary” or junk foods [6].

### 5.2. Moderating Variables

The meta-analysis revealed that the timing of food measurement (immediate versus delayed) played a critical role in understanding the effects of television on food intake. Specifically, when we analysed the effect of television on later meal intake, television viewing consistently and strongly emerged as a factor that increased food intake. Beyond the reduced energy expenditure associated with television viewing, it has been argued that distractions such as televisions can impact eating behaviour by reducing awareness of one’s eating behaviours [50], impairing the awareness and integration of physiological signals of satiation and satiety [51], and impairing mnemonic processes required to regulate long-term food intake [49,50]. For example, one study included in this review showed that participants eating while watching TV ate more cookies at a later meal and reported a less vivid memory of the meal compared to the control group [30]. This led the authors to hypothesise that participants were paying less attention to their meals, leading to impairment in the encoding of the meal and poorer memory for the meal later. Indeed, in another study, for the participants in the television group, the memory recall of prior intake was poor, and the consequent food intake was higher [13]. Conversely, if distraction from television during a meal increases later eating, then improving memory for lunch should work in the opposite direction. Indeed, later studies showed that increasing awareness of food, as covered in previous reviews [3]. For example, Higgs and Donohoe showed that focusing on food during lunch enhances memory for one’s lunch and reduces afternoon snack intake [52]. Indeed, the importance of attention and mnemonic processes in regulating food intake has been considered by other meta-analyses on distraction and food intake [3]. Taken together, studies in this review show that environmental factors that impact the encoding of information about a recent eating episode (e.g., distractions such as TV) may contribute to overeating and possibly weight gain.

### 5.3. Key Considerations

An important strength of this systematic review and meta-analysis was the focus on the experimental manipulation of television watching on food intake. Focusing on experimental studies ensures rigorous experimental control (e.g., random allocation, controlled lab conditions) over confounds that could impact the interpretation of results. Indeed, removing studies that included additional confounds (advertising, toys, classroom setting) reduced heterogeneity in the effect sizes and allowed a clearer picture to resonate from the literature. This review, therefore, can make confident conclusions about the impact of television viewing on food intake.

One consideration is that the “control” group in these experimental studies typically involved no television and, indeed, no other distractors. This has two important consequences. First, it brings into question the ecological validity of experiments where participants are compelled to display unnatural behaviours (e.g., eating and doing nothing). Second, in the absence of any external stimulation (e.g., TV, phone, social conversations), as one would likely encounter in a real-world setting, additional focus could be placed on the food itself. This could facilitate eating behaviour, resulting in increased food intake in a manner similar to the television-watching manipulation. Therefore, when you compare intake between the two groups, there is little difference as both show increased intake. The fact that television viewing impacts later meal intake suggests these concerns are not problematic. In fact, it points to the mechanism by which television impacts energy intake via psychological factors involving memory processes.

## 6. Conclusions

This systematic and meta-analytic review provides a focused synthesis of the experimental studies investigating whether watching television while eating leads to overeating. While the overall findings suggested mixed results, delayed food intake was more strongly influenced by television viewing. These findings highlight the complexity of eating behaviours and underscore the importance of considering both immediate and long-term impacts when studying the relationship between external conditions and food intake. The findings have implications for understanding the role of external distractions (e.g., television) on diet and could inform strategies to promote healthier eating behaviours in different populations. Understanding the role of psychological factors in overeating is critical to developing effective interventions (e.g., mindfulness to increase attention or cognitive training to enhance working and episodic memory) to reduce overeating and curb the rise of obesity.

## Figures and Tables

**Figure 1 nutrients-17-00166-f001:**
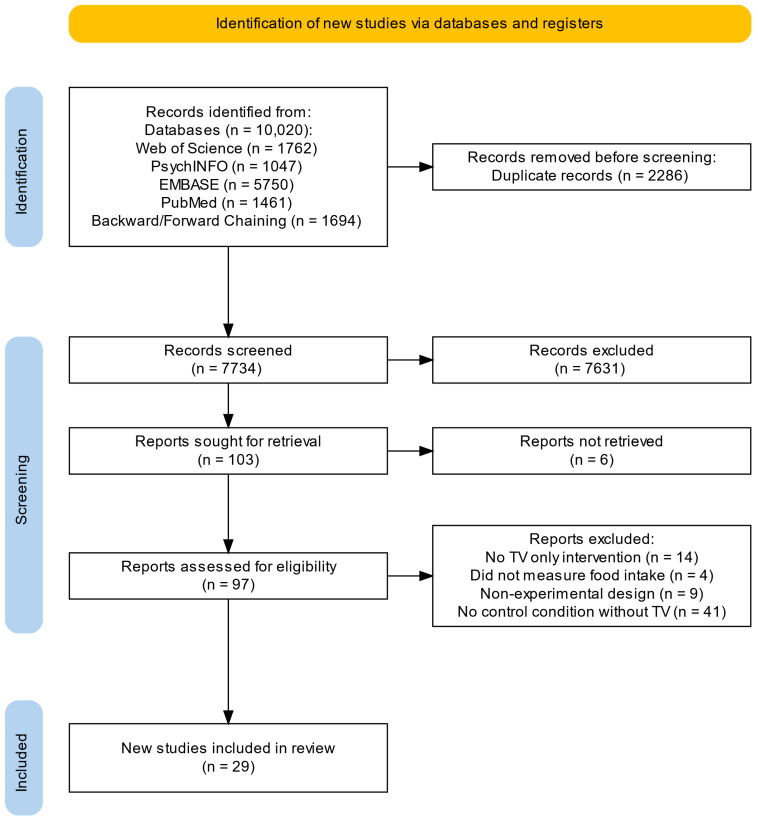
PRISMA flowchart for study search and screening.

**Figure 2 nutrients-17-00166-f002:**
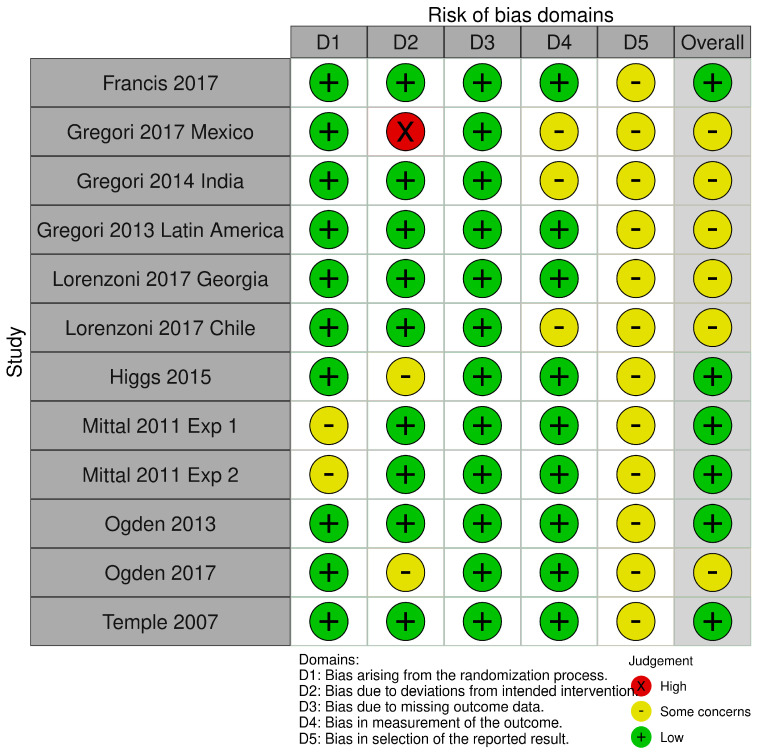
Risk of bias traffic light plot for parallel RCTs [13,35,36,37,38,39,41,42,44,45,46].

**Figure 3 nutrients-17-00166-f003:**
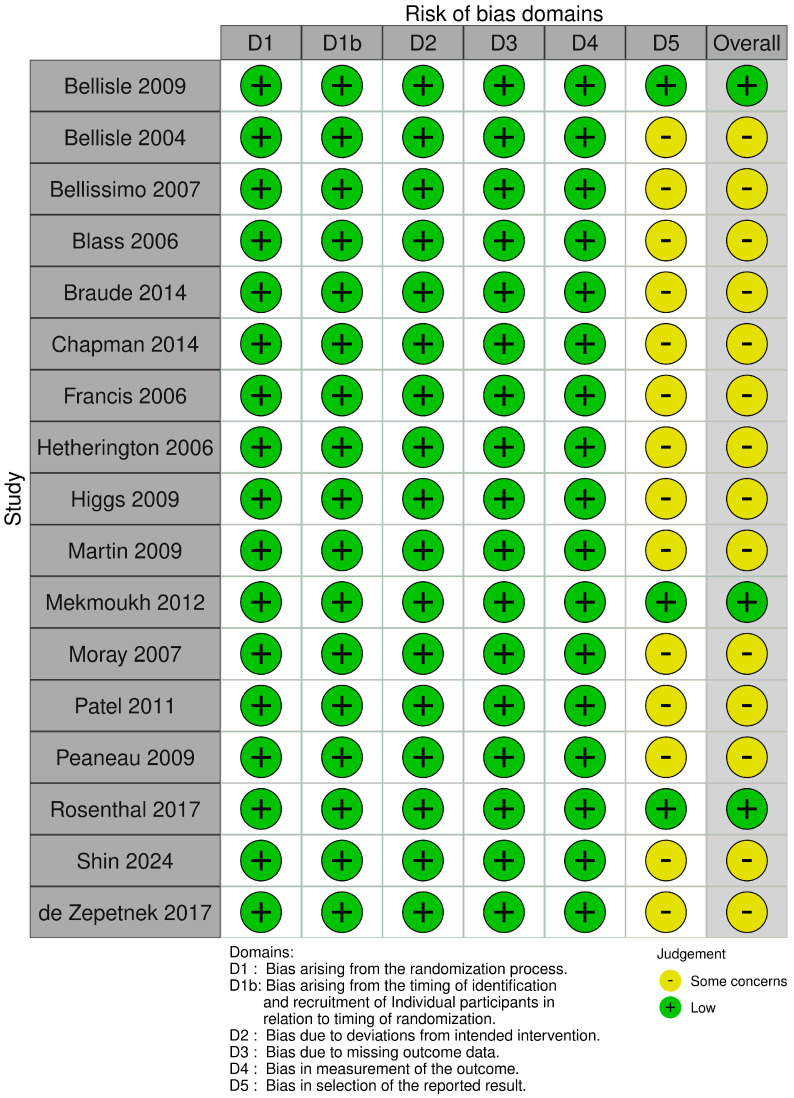
Risk of bias traffic light plot for cross-over RCTs [23,24,25,26,27,28,29,30,31,32,33,34,40,43,47,48,49].

**Figure 4 nutrients-17-00166-f004:**
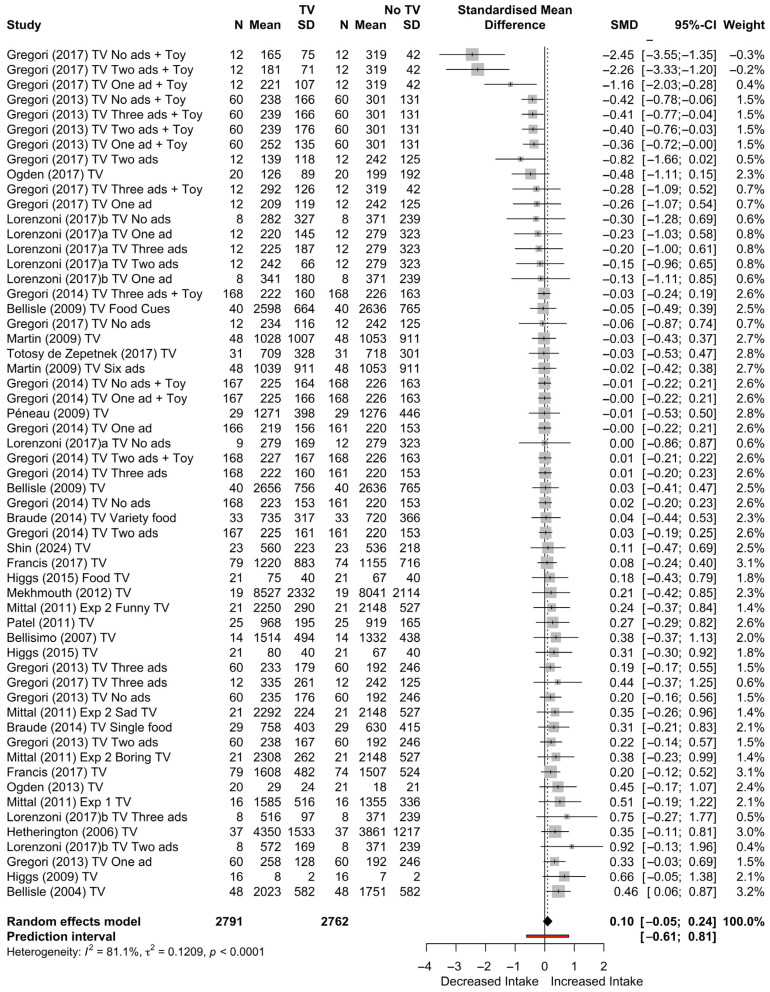
Forest plot of k effect sizes across all studies [13,23,24,25,26,27,28,29,30,31,32,33,34,35,36,37,38,39,40,41,42,43,44,45,46,47,48,49].

**Figure 5 nutrients-17-00166-f005:**
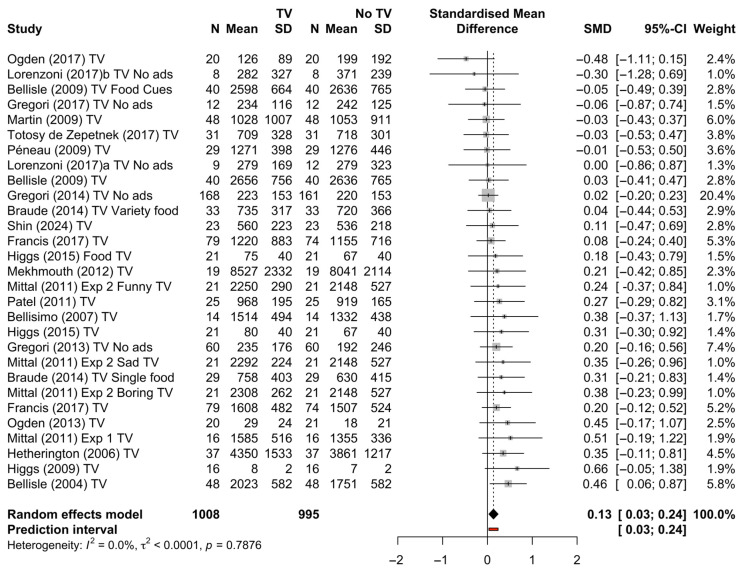
A forest plot of the study effect sizes for television manipulations only [13,23,25,27,29,30,31,32,33,34,35,36,37,38,41,42,45,46,47,48,49].

**Figure 6 nutrients-17-00166-f006:**
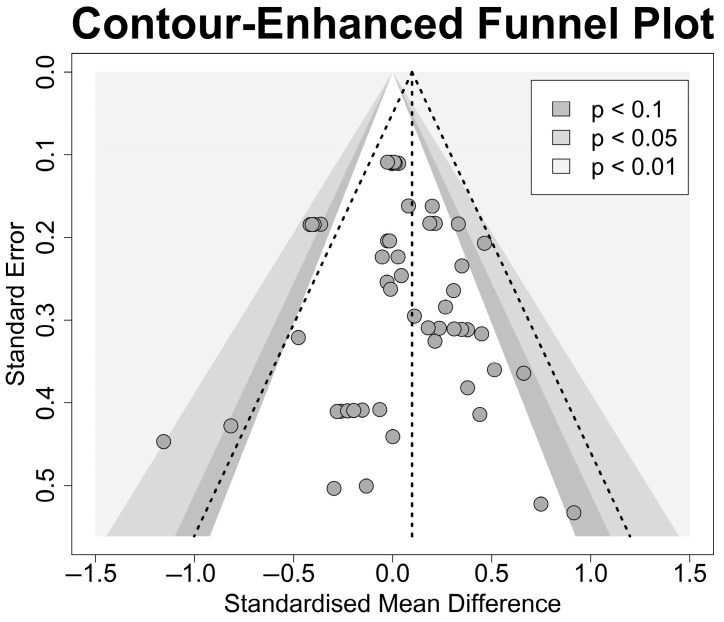
Funnel plot of effect sizes and standard errors.

**Figure 7 nutrients-17-00166-f007:**
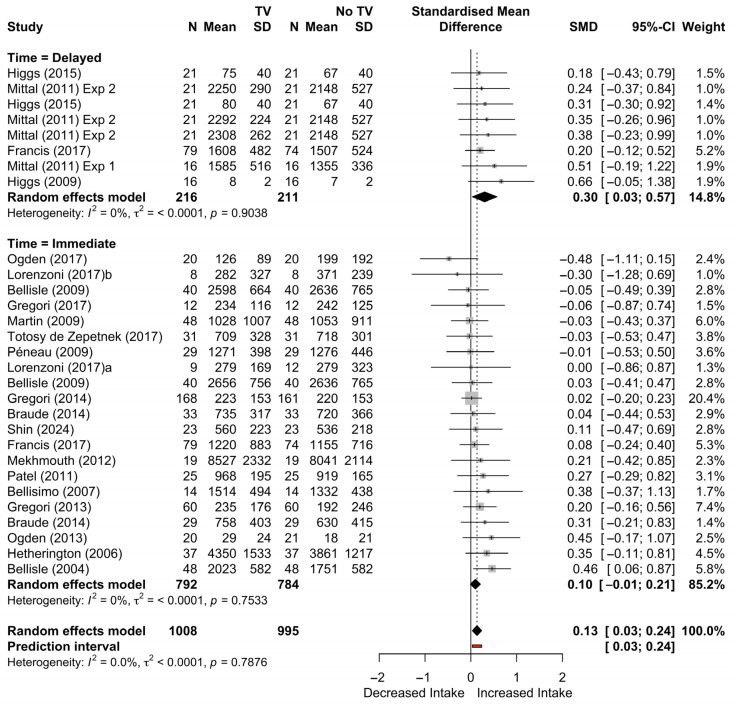
Forest plot of k effect sizes by intake timing (delayed [**top**] or immediate [**bottom**]) for television only studies [13,23,25,27,29,30,31,32,33,34,35,36,37,38,41,42,45,46,47,48,49].

**Table 1 nutrients-17-00166-t001:** Key findings for 29 articles analysed.

Authors, Year	Country	Participant Population [N, % Females, Age, BMI]	Study Design	Experimental Conditions	Participants per Group	FI Measure	Key Findings	Confounds
Péneau et al. (2009) [34]	France	n = 29 48.3% females Age 15.5 BMI 21.3 ± 0.45	Within-subjects design	4 conditions (eating alone, eating in groups of 3, eating with TV, eating with music)	29 participants per group	kJ	Higher intake of solid foods and liquids in males than in females.	Social inhibition was viewed in teenagers; music may affect FI.
Mittal et al. (2019) [13] Experiment 1	Australia	n = 32 100% females Age 20.55 ± 3.85 BMI 21.65 ± 1.85	Between-subjects design	2 conditions: (snacking with/without TV)	16 participants per group	kJ	Participants who viewed while snacking consumed more food later in the test meal and were poorer at recalling the amount of food consumed.	Prior consumption of snacks may act as a confounder during the test meal.
Mittal et al. (2019) [13] Experiment 2	Australia	n = 32 100% females Age 20.55 ± 3.85 BMI 21.65 ± 1.85	Between-subjects design	4 conditions: (snacking without TV, snacking with boring, sad or funny TV)	21 participants per group	kJ	No overall significant differences between conditions. Greater FI in boring and funny conditions relative to no TV control.	Presence of food cues in each TV condition is not controlled.
Bellisle et al. (2009) [48]	France	n = 40 100% females Age 26.4 (low restraint), 25.9 (high restraint) BMI 21.5 (low restraint), 22.4 (high restraint)).	Conditions as within-subjects and restraint as between-subject design	5 conditions: eating alone, eating in groups of 3, eating with food ad, eating with no food ad, eating with audio	40 participants per group (20 high restraint, 20 low restraint)	kJ	Participants consumed less food when eating alone than in other conditions. No other significant within-group differences were observed.	Levels of restraint: HR women were more affected by distractors than LR women.
Lorenzoni et al. (2017)a [39]	Chile	n = 40 100% females Age 6–12 BMI 21.08	Between subjects-design.	5 conditions: no TV exposure, TV with no ads, TV with one ad, TV with two ads, TV with three ads	8 participants per group	kcal	TV and advertisement exposure did not affect energy intake.	The content of advertisements may act as a confounder.
de Zepetnek et al. (2017) [27]	Canada	n = 31 100% females Age 11.7 ± 0.3 BMI 49.3 ± 3.1.	Within-subjects design	4 conditions: exposure to TV, video game playing, computer tasks, no screen exposure (control)	31 participants per group	kcal	Pre-meal exposure to screen did not affect FI.	Video games and computer tasks may act as confounders.
Odgen et al. (2013) [41]	England	n = 81 100% females Age 22 ± 5.18 BMI 22.7 ± 5.16	Between-subjects design	4 conditions: exposure to TV, driving, social interaction, eating alone	Driving: 21, Television: 20, Social: 19, Alone: 21	g	Participants viewing TV consumed the most food, suggesting that television viewing has a greater impact than other distractors.	Social interaction and driving may act as confounders.
Chapman et al. (2014) [26]	Sweden	n = 18 100% females Age 22 ± 1.3 BMI 21.1 ± 1.1	Within-subjects design	3 conditions: boring TV vs. engaging TV vs. text (control)	18 participants per group	g	FI was the highest in boring TV condition and the lowest in engaging TV condition.	Text conditions may act as a confounder; the content of the TV has subjective effects.
Braude & Stevenson (2014) [49]	Australia	n = 62 100% females Age 19.6 ± 2.2 BMI (Single food) 21.9 ± 2.0 BMI (Variety) 22.3 ± 2.7	Two-way mixed design	Between factor condition: single food vs. multiple food. Within subjects condition: television vs. no television	Single food: 29 Variety food: 33	kJ	Participants exposed to TV viewing consumed a greater amount of food.	Multiple snacks may act as confounders.
Gregori et al. (2014) [36]	India	n = 1680 50% females Age 6.5 BMI not reported	2 × 5 full factorial design.	5 conditions: no exposure to TV, TV without ads, TV with 1 ad, TV with 2 ads, TV with 3 ads 2 conditions: toy vs. no toy	168 participants per group	kcal	No significant difference was found in the FI between various television conditions. Gadgets did not affect FI.	Exposure to a gadget (toy) may act as a confounder.
Patel et al. (2011) [33]	Canada	n = 25 100% females Age 11.5 ± 0.4 BMI percentile 55.8 ± 5.2	Within-subjects design	2 conditions: exposure to TV vs. no exposure to TV	25 participants per group	kcal	Television viewing did not have an effect on FI. Post-glucose FI was lower in post-pubertal girls than in peri-pubertal girls.	Premeal glucose drink may act as a confounder.
Odgen et al. (2017) [42]	England	n = 60 100% females Age 24.0 ± 3.3 BMI 22.4 ± 3.5	Between-subjects design	3 conditions: exposure to television, walking, social interaction	20 participants per group	kcal	There were no overall differences between the conditions. Participants with restrained eating, however, ate more while walking than watching TV.	The consumption of a cereal bar may act as a confounder.
Temple et al. (2007) [44]	USA	n = 26 53% females Age 11.2 ± 1.2 BMI 18.2 ± 2.3	Between-subjects design	3 conditions: continuous TV show, repeated segment of a TV show, no television	Control = 9 Continuous = 9 Repeated = 8	kcal	Continuous TV viewing results in the highest FI. Allocating attention to TV disrupts habituation to food cues.	The content of the TV in both TV conditions may act as a confounder.
Francis et al. (2017) [35]	Australia	n = 153 62% females Age 19.7 ± 2.9 BMI 22.4 ± 3.1	Between-subjects design	2 conditions: snack intake with TV vs. without TV	No TV condition: 74 participants. TV condition: 79 participants	kJ	Women consumed more snacks with TV, while men consumed more without TV. Men who snacked ate more lunch than their counterparts.	Snack intake may act as a confounder for lunch intake.
Lorenzoni et al. (2017)b [38]	Georgia	n = 57 50% females Age 6 BMI not reported	Between-subjects design	5 conditions: no TV exposure, TV with no ads, TV with one ad, TV with two ads, TV with three ads	Every group had 12 participants; TV without ad condition had 9	kcal	TV and advertisement exposure did not affect energy intake in children.	Advertisement content may act as a potential confounder.
Bellisle et al. (2004) [25]	France	n = 48 100% females Age 29.9 ± 1.4 BMI 22.3 ± 0.2	Within-subjects design	4 conditions: baseline lunch, exposure to TV, exposure to audio, last lunch	48 participants per group	kcal	Food consumption in both TV and audio conditions was higher than in control conditions.	Audio conditions may act as a confounder; strict lab settings present do not generalise into real living.
Higgs & Woodward (2009) [30]	England	n = 16 100% females Age 19 ± 1 BMI 21.7 ± 1.75	Within-subjects design	2 conditions: exposure to TV, control (no exposure to TV)	16 participants per group	g	Participants who ate lunch in the TV condition consumed more subsequent snacks and had a poorer recall of the lunch consumed.	The time gap between lunch and cookie intake may be a confounder.
Bellisimo et al. (2007) [47]	Canada	n = 14 0% females Age 12.6 ± 0.4 BMI 20.4 ± 0.9	2 × 2 factorial within-subjects design.	2 conditions: exposure to TV vs. no TV and glucose and Splenda intake	14 participants per group	kcal	FI was higher in the TV condition. Exposure to TV overrides the satiety signals in boys.	Prior intake of glucose and Splenda may affect FI.
Mekhmouth et al. (2012) [32]	France	n = 38 0% females Age 16.45 ± 0.95 BMI 25.3 ± 2.7	Between-subjects design	4 conditions: alone, group, television, music	Data Unavailable	kJ	FI for normal weight boys was higher in music condition and for overweight boys in TV condition.	Social interaction and music may act as confounders.
Rosenthal & Raynor (2016) [43]	USA	n = 20 85% females Age 22.3 ± 3.7 BMI 21.6 ± 2.3	Between-subjects design.	2 conditions: exposure to TV vs. no TV	10 participants per group	g/kcal	Exposure to TV did not produce significant effects on FI.	Confounders such as portion sizes and a fixed time to eat were present.
Hetherington et al. (2005) [29]	England	n = 37 43.2% females Age 28.3 ± 1.7 BMI 23.9 ± 0.8	Within-subjects design	4 conditions: eating alone, TV exposure, strangers, friends	37 participants per group	kJ	FI was higher when consumed with TV or friends than in other groups.	Social interaction may affect FI.
Martin et al. (2009) [31]	USA	n = 48 54% females Age 31.9 ± 1.5 BMI 25.8 ± 0.6	Within-subjects design	4 conditions: control, reading, TV with no ads, TV with ads	48 participants per group	kcal	TV did not affect energy intake.	Reading and advertisements may affect FI.
Blass et al. (2006) [24]	USA	n = 20 75% females Age not reported BMI 24.5 ± 5.34	Within-subjects design	2 conditions: exposure to TV, listening to audio	20 participants per group	kcal	FI was higher during the TV condition.	Music and a variety of target foods may affect FI.
Gregori et al. (2017) [37]	Mexico	n = 120 50% females Age 6.5 BMI 16.40	2 × 5 full factorial design	5 conditions: no exposure to TV, TV without ads, TV with 1 ad, TV with 2 ads, TV with 3 ads. 2 conditions: toy vs. no toy	12 participants per group	kcal	Exposure to TV and ads did not affect the FI in participants.	The presence of toys and TV ads may affect FI.
Moray et al. (2007) [40]	USA	n = 20 50% females Age 20.8 BMI not reported	Within-subjects design	2 conditions: FI with and without television	20 participants per group	g	FI was higher in the television group due to impaired attention to food consumption.	Ceiling effect of the food provided may affect FI.
Francis et al. (2006) [28]	USA	n = 24 50% females Age 4.6 ± 0.7 BMI 15.9 ± 1.2	2 × 2 factorial design within-subjects design	2 conditions: lunch intake with or without television	24 participants per group	g	FI in television condition was significantly more.	Daily TV-watching habits at home may have an effect on FI.
Gregori et al. (2013) [46]	Latin America	n = 600 50% females Age 6.5 BMI 16.38	2 × 5 full factorial design	5 conditions: no exposure to TV, TV without ads, TV with 1 ad, TV with 2 ads, TV with 3 ads. 2 conditions: toy vs. no toy	60 participants per group	kcal	Exposure to TV and ads did not affect the FI in participants.	The presence of toys and TV ads may affect FI.
Higgs (2015) [45]	England	Study 2 n = 63 100% females Age 19.7 ± 3.5 BMI 22.1 ± 3.4	Between-subjects design	3 conditions: TV food, TV, control	21 participants per group	g	Food intake was the highest when TV was present.	-
Shin (2024) [23]	Republic of Korea	n = 23 53.33% females Age 24.96 ± 1.36 BMI 23.08 ± 0.93	Within-subjects design	4 conditions: audio, TV, Smartphone and control	23 participants per group	g	Intake was the highest when using smartphones. Satiety was highest in the control setting.	Participants’ consumption of food and drinks before the experiment may affect the results.

**Table 2 nutrients-17-00166-t002:** Moderation of k effect sizes.

Moderators	k	F-Statistic	*p*-Value	B	t-Statistic	*p*-Value	Lower CI	Upper CI
**Timing**	29	3.38	0.08					
*Immediate*	21			0.098	1.84	0.077	−0.023	0.427
*Delayed*	8			0.300	3.08	0.005 *	0.100	0.501
**BMI**	23	0.20	0.66	−0.010	−0.448	0.659	−0.059	0.038
**Age**	28	1.22	0.28	0.006	1.10	0.279	−0.005	0.018
**Sex**	29	1.21	0.28	0.002	1.10	0.280	−0.002	0.005
**Study design**	29	0.38	0.69					
*Between*	14			0.125	2.10	0.045 *	0.002	0.247
*Mixed*	3			−0.032	−0.251	0.804	−0.296	0.231
*Within-subjects*	12			0.077	0.738	0.467	−0.137	0.291
**Risk of bias**	29	0.55	0.46					
*Low*	9			0.183	2.55	0.017 *	0.036	0.329
*Some concerns*	20			0.114	−0.743	0.464	−0.257	0.120

* *p* < 0.05.

## Data Availability

The raw data and code supporting the conclusions of this article will be made available by the authors upon request.

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
