# Peer review of "Watching Television While Eating Increases Food Intake: A Systematic Review and Meta-Analysis of Experimental Studies"

_nutrients, 2025, doi:10.3390/nu17010166_

Round 1

Reviewer 1 Report

Comments and Suggestions for Authors

Dear Authors,

The reviewed manuscript meets the requirements of a correct scientific paper. The main strength is the efficient use of analytical techniques - but for a final small set of studies (29). Actually, in the Introduction (and in the Abstract), based on the literature review, the authors presented the links between television viewing and weight gain and obesity. These correlations were then confirmed on the basis of the analyses conducted. Thus, the manuscript does not contribute new knowledge regarding the issue under study. In my opinion, the statement in the Conclusions that distraction (TV viewing) affects mnemonic processes does not come from the analyses performed. This result comes from only one article included in the analyses (ref. 49) and another used in the Discussion of Results (ref. 50). I also think that the original added value of the manuscript would be to suggest intervention tools, which the authors wrote about in the Conclusions.

I have two minor suggestions: 1. Nutrients uses a different way of citing references, and 2. Table 1 does not indicate the country for references 42 and 23.   

Kind regards

Author Response

Comment 1: In my opinion, the statement in the Conclusions that distraction (TV viewing) affects mnemonic processes does not come from the analyses performed. This result comes from only one article included in the analyses (ref. 49) and another used in the Discussion of Results (ref. 50).

Response 1: We agree broadly with this statement - the analysis itself only indicates that the experimental studies show that television impacts food intake at a later meal, rather than immediately. Mnemonic processes were put forward as an interpretation to understand why we observed the pattern of results in the meta-analyses. We have made this point clear in the main Discussion and removed it from the Abstract.

Comment 2: I also think that the original added value of the manuscript would be to suggest intervention tools, which the authors wrote about in the Conclusions.

Response 2: Thank you for this suggestion. We have included examples of potential intervention tools based on evidence provided in our analysis.

Comment 3: 1. Nutrients uses a different way of citing references

Response 3: The in-text citations and references section have been updated to match the formatting requirements.

Comment 4: 2. Table 1 does not indicate the country for references 42 and 23.

Response 4: The countries for these references have been added. Thank you for picking up on this missing information.

Reviewer 2 Report

Comments and Suggestions for Authors

The manuscript entitled “The effect of viewing television on energy intake: A systematic review and meta-analysis” presents interesting issues; however, certain revisions should be made.

The topic concerns television viewing and energy intake. Although the results are negative, it is still a valuable study. I wonder if it would be a good idea to modify the title of the article based on the obtained results – it might contribute to better citation of the manuscript.

The manuscript needs to be adjusted to the guidelines of this journal and formatted accordingly (e.g., text alignment).

Among these 29 publications, there is a part about children and adolescents – it would be worthwhile to conduct these analyses separately.

A total of 23 publications were included in the meta-analysis – please provide the criteria for their selection.

The conclusions should be written more clearly.

The conclusions in the abstract emphasize that the conducted study is not necessary and there is a lack of emphasis on the novelty of these findings.

A systematic review should also include an assessment of the quality of reporting articles (e.g. RoB 2 Tool)

Author Response

Comment 1: The topic concerns television viewing and energy intake. Although the results are negative, it is still a valuable study. I wonder if it would be a good idea to modify the title of the article based on the obtained results – it might contribute to better citation of the manuscript.

Response 1: We have updated the title to reflect both the novelty and value of the review and meta-analysis.

Comment 2: The manuscript needs to be adjusted to the guidelines of this journal and formatted accordingly (e.g., text alignment).

Response 2: The in-text citations and references section and general formatting have been updated to match the formatting requirements of Nutrients.

Comment 3: Among these 29 publications, there is a part about children and adolescents – it would be worthwhile to conduct these analyses separately.

Response 3: Thank you for this suggestion. We included age as a continuous variable in moderation analyses and this did not change the pattern of results. Conducting your suggested analysis splitting the sample into children (<13), adolescents and adults (18+) also did not change the pattern of results. It should also be noted that only 2 studies included an adolescent/pubescent sample so analysis is limited. Given this did not change results and the overlap with the already included moderation analysis on age as a continuous variable, we have opted not to include this additional analysis on the child/adolescent split and keep age as a continuous variable in the moderation analysis.

Comment 4: A total of 23 publications were included in the meta-analysis – please provide the criteria for their selection.

Response 4: The articles that included enough information to use for meta-analysis were included. Authors were also contacted to request additional data not otherwise reported. The criteria for inclusion for meta-analysis has been added to the manuscript.

Comment 5: The conclusions should be written more clearly.

Response 5: We have updated the conclusions to focus on the conclusions brought about directly from the review and meta-analyses.

Comment 6: The conclusions in the abstract emphasize that the conducted study is not necessary and there is a lack of emphasis on the novelty of these findings.

Response 6: Thank you for pointing this out. We have updated the abstract to emphasise the novelty and importance of the findings from the meta-analyses.

Comment 7: A systematic review should also include an assessment of the quality of reporting articles (e.g. RoB 2 Tool)

Response 7: We agree with this comment. We have included the Cochrane Risk of Bias-2 quality assessment tool for both parallel (between-subjects) and cross-over (within-subjects) trials as a measure of study quality. We have also included risk of bias as an additional potential moderator and added this into Table 2.